# Molecular Characterization of Portuguese Patients with Hereditary Cerebellar Ataxia

**DOI:** 10.3390/cells11060981

**Published:** 2022-03-12

**Authors:** Mariana Santos, Joana Damásio, Susana Carmona, João Luís Neto, Nadia Dehghani, Leonor Correia Guedes, Clara Barbot, José Barros, José Brás, Jorge Sequeiros, Rita Guerreiro

**Affiliations:** 1UnIGENe, IBMC-Institute for Molecular and Cell Biology, i3S-Instituto de Investigação e Inovação em Saúde, Universidade do Porto, 4200-135 Porto, Portugal; jlnetosci@gmail.com (J.L.N.); jorge.sequeiros@ibmc.up.pt (J.S.); 2Neurology Department, Centro Hospitalar Universitário do Porto, 4099-001 Porto, Portugal; josebarros@chporto.min-saude.pt; 3CGPP-Centre for Predictive and Preventive Genetics, IBMC-Institute for Molecular and Cell Biology, i3S-Instituto de Investigação e Inovação em Saúde, Universidade do Porto, 4200-135 Porto, Portugal; clarabarbot@gmail.com; 4Department of Neurodegenerative Science, Van Andel Institute, Grand Rapids, MI 49503, USA; susanammcarmona@gmail.com (S.C.); nadia.dehghani@vai.org (N.D.); jose.bras@vai.org (J.B.); rita.guerreiro@vai.org (R.G.); 5Department of Neurosciences and Mental Health, Neurology, Hospital de Santa Maria, Centro Hospitalar Universitário Lisboa Norte, 1649-028 Lisbon, Portugal; lcorreiaguedes@gmail.com; 6Instituto de Medicina Molecular João Lobo Antunes, Faculdade de Medicina, Universidade de Lisboa, 1649-028 Lisbon, Portugal; 7CNS-Campus Neurológico Sénior, 2560-280 Torres Vedras, Portugal; 8ICBAS-School of Medicine and Biomedical Sciences, Universidade do Porto, 4050-313 Porto, Portugal; 9Division of Psychiatry and Behavioral Medicine, Michigan State University College of Human Medicine, Grand Rapids, MI 49503, USA

**Keywords:** cerebellar ataxia, recessive ataxia, exome sequencing, molecular mechanisms, de novo variant

## Abstract

Hereditary cerebellar ataxia (HCA) comprises a clinical and genetic heterogeneous group of neurodegenerative disorders characterized by incoordination of movement, speech, and unsteady gait. In this study, we performed whole-exome sequencing (WES) in 19 families with HCA and presumed autosomal recessive (AR) inheritance, to identify the causal genes. A phenotypic classification was performed, considering the main clinical syndromes: spastic ataxia, ataxia and neuropathy, ataxia and oculomotor apraxia (AOA), ataxia and dystonia, and ataxia with cognitive impairment. The most frequent causal genes were associated with spastic ataxia (*SACS* and *KIF1C*) and with ataxia and neuropathy or AOA (*PNKP*). We also identified three families with autosomal dominant (AD) forms arising from de novo variants in *KIF1A*, *CACNA1A,* or *ATP1A3*, reinforcing the importance of differential diagnosis (AR vs. AD forms) in families with only one affected member. Moreover, 10 novel causal-variants were identified, and the detrimental effect of two splice-site variants confirmed through functional assays. Finally, by reviewing the molecular mechanisms, we speculated that regulation of cytoskeleton function might be impaired in spastic ataxia, whereas DNA repair is clearly associated with AOA. In conclusion, our study provided a genetic diagnosis for HCA families and proposed common molecular pathways underlying cerebellar neurodegeneration.

## 1. Introduction

Hereditary cerebellar ataxias (HCA) comprise a heterogeneous group of neurological disorders. The cardinal cerebellar syndrome of HCA may be present in isolation or accompanied by a wide constellation of neurological and non-neurological symptoms and signs. All modes of inheritance have been recognized, with the autosomal recessive (AR) forms being the most complex, both from a clinical and genetic point of view [1]. Prevalence of autosomal dominant (AD) HCA ranges from 0.0 to 5.6/100,000, and that of AR-HCA varies from 0.0 to 7.2/100,000 [2]. AD-HCA are generally caused by repeat expansions, with Machado-Joseph Disease/Spinocerebellar ataxia type 3 (MJD/SCA3) being, overall, the most frequent form and SCA2 the second on a global basis, but first in countries such as Cuba or India [2,3]. Friedreich ataxia is the most frequent recessive ataxia, followed by ataxia telangiectasia or ataxia with oculomotor apraxia (AOA), depending on the series [2,3]. The genetic diversity of HCA has greatly increased over the past years, due to the contribution of next generation sequencing (NGS) techniques. With a diagnostic yield of 20–25%, NGS has allowed for the identification of new ataxia-causing genes and redefining genotype–phenotype correlations; however, it has uncovered a large number of variants of unknown clinical significance [4,5].

Despite the diversity of HCA-genes and their ubiquitous expression throughout the brain, genetic defects in HCA predominantly affect the cerebellum, suggesting that cerebellar cells (particularly, Purkinje cells) are more vulnerable. Therefore, the identification of shared molecular pathways between mutated genes is crucial to understand the basis of cerebellar neurodegeneration [6,7]. Many genes function in common pathways, including protein quality control, transcription regulation, and calcium homeostasis in AD-HCA [8]; and DNA repair, mitochondria function, and lipid metabolism in AR-HCA [9]. Based on common pathways identified, it may be possible to develop effective therapeutic strategies, targeting more than one form of HCA [7].

This study focused on 19 Portuguese families with apparent AR-HCA, previously identified during a large population-based survey and studied by targeted testing for the most common genes [10], and now investigated using whole-exome sequencing (WES). The molecular mechanisms potentially responsible for the neurodegeneration associated with the identified HCA causal-genes were reviewed and discussed.

## 2. Materials and Methods

### 2.1. Patients and Clinical Study

A national, systematic, population-based survey was conducted in Portugal from 1994 until 2004, aiming at identifying Portuguese families with HCA and hereditary spastic paraplegia (HSP) [10]. Its detailed methodology was previously described [10]; briefly, clinical and family history, neurological evaluation and blood sampling were performed, and genomic DNA extracted. Written informed consent was obtained from participants.

### 2.2. Genetic Analysis

In the national survey, 250 families (344 patients) with presumed autosomal recessive ataxia were identified. Targeted genetic testing was conducted ab initium, according to the main clinical phenotype; 61 families (83 patients), however, remained without a genetic diagnosis [10]. Pathogenic variants in *PNKP* [11] were later identified in eight families (11 patients), and in *MAG* [12] in one family (3 patients), all with AOA. More recently, expansion of an intronic repeat in *RFC1* [13] was excluded in all families with genomic DNA available who persisted without diagnosis.

From the 61 undiagnosed families, we only had genomic DNA available for 27 families, with a final diagnosis being reached in the 19 here described. We performed whole-exome sequencing (WES) in at least one patient of each family (depending on the availability of genomic DNA). Exome sequencing libraries were prepared using a SureSelect Exome Capture Kit v7 (Agilent), and sequencing was performed on NextSeq550 with 100 bp paired-end reads, according to the manufacturer’s instructions. After sequencing, reads were then aligned to the human reference genome hg19/GRCh37 using Burrows-Wheeler Aligner (bwa) v0.7.1 [14] and variants were called using GATK best practices v3.3-0 [15]. Duplicate reads were identified through samblaster v0.1.21 [16]. Variant annotation was performed with snpEff v4.2 [17] and dbNSFP v2.9 [18].

Annotated variants were filtered to include rare variants with a minor allele frequency (MAF) <1% in population databases such as Nucleotide Polymorphism Database (dbSNP; https://www.ncbi.nlm.nih.gov/snp (accessed on 11 February 2022)), Genome Aggregation Database (gnomAD v2.1; https://gnomad.broadinstitute.org (accessed on 11 February 2022)), and the 1000 Genome Project ((http://www.1000genomes.org (accessed on 11 February 2022)). Non-synonymous and splice-site variants, and variants for AR forms were filtered for primary analysis. This analysis was performed using Exomiser v7.2.1 [19], which also allowed variant prioritization based on the human phenotype ontology HP:0001251 (term name:ataxia). Then, we excluded intronic, UTR, intergenic, and synonymous variants, as well as homozygous variants reported in gnomAD. In addition, when de novo variants were suspected, those for AD forms were also filtered and analyzed. The functional predicted impact of missense variants was evaluated using SIFT [20], Mutation Taster [21], Polyphen2 [22], Mutation assessor [23], FATHMM [24], and UMD-Predictor [25]. Analysis of splice-site variants was performed using Splice Site Finder-like [26], MaxEntScan [27], Neural Network SPLICE (NNSPLICE) v0.9 [28], GeneSplicer [29], and Human Splicing Finder (HSF) v3.1 [30]. Variants were classified according to the ACMG/AMP Standards and Guidelines [31]. ClinVar (http://www.ncbi.nlm.nih.gov/clinvar (accessed on 11 February 2022)) and Human Gene Mutation Database (http://www.hgmd.cf.ac.uk/ac/index.php (accessed on 11 February 2022)) were also used for variant classification.

Relevant variants identified by WES were confirmed by Sanger sequencing, which also allowed verifying intrafamilial segregation in several families. For PCR amplifications, we used Ranger Mix (Bioline, London, UK), purified products with Exo/SAP (GRiSP, Porto, Portugal), and performed Sanger sequencing using Big Dye Terminator Cycle Sequencing v1.1 (Applied Biosystems, Foster City, CA, USA) in an ABI 3130xl Genetic Analyzer (Applied Biosystems, Foster City, CA, USA). Sequencing analysis was carried out using the SeqScape v2.6 software (Applied Biosystems, Foster City, CA, USA).

### 2.3. Minigene Assays

The *SPG11* and *KIF1C* minigenes constructs were obtained by cloning the exonic and intronic sequences flanking the variants of interest (c.3039-5T > G in *SPG11* and c.1166-2A > G in *KIF1C*) into the pCMVdi vector, kindly provided by Dr Alexandra Moreira [32]. Briefly, *SPG11* exons 16, 17, and 18 with intronic regions or *KIF1C* exons 13, 14, and 15 with intronic regions were PCR amplified from patients’ genomic DNA and cloned into the pCMVdi vector, using the Gibson assembly method. Sequences were modified by site-directed mutagenesis to generate the wild-type constructs, using a QuikChange II Kit (Agilent, Santa Clara, CA, USA), according to the manufacturer’s protocol.

HEK293T cells were transfected with the minigene constructs using jetPRIME (Polyplus-transfection, Illkirch, France), according to the manufacturer’s protocol. RNA was extracted 48 h after transfection, using NZYol (Nzytech, Lisbon, Portugal), as per manufacturer’s recommendations, followed by purification of the RNA aqueous phase, using an RNeasy mini kit (Qiagen, Hilden, Germany). RNA quantification was performed on NanoDrop 2000 (ThermoFisher Scientific, Waltham, MA, USA). cDNA was synthesized by reverse transcription-PCR of 2 µg total RNA with oligo (dT), using a SuperScript III first-strand synthesis system (Invitrogen, Carlsbad, CA, USA), according to the manufacturer’s protocol. The resulting cDNA was amplified by PCR and loaded on agarose gel for extraction with a Zymoclean Gel DNA Recovery Kit (Zymo Research, Irvine, CA, USA). The resulting products were sequenced by Sanger sequencing using Big Dye Terminator Cycle Sequencing v1.1 (Applied Biosystems, Foster City, CA, USA) in an ABI 3130xl Genetic Analyzer (Applied Biosystems, Foster City, CA, USA).

## 3. Results

### 3.1. Genetic and Molecular Characterization

We used WES to identify variants and genes causing HCA in 19 Portuguese families with family history compatible with AR transmission (including 9 with known consanguinity), in a total of 30 individuals: 19 index cases, one affected and 10 non-affected relatives. At least, one relative of seven index cases was available. All relevant variants (rare, pathogenic, likely pathogenic, and/or predicted to be deleterious) were confirmed by Sanger sequencing, and segregation analysis was performed with all available family members. Phenotypic classification considered the main clinical symptom associated with ataxia: spasticity, neuropathy, oculomotor apraxia, dystonia, or cognitive impairment.

Overall, we identified 24 rare nucleotide variants in 13 genes (Appendix A): *SACS*, *KIF1C*, *ANO10*, *SPG11*, *SYNE1,* and *CACNA1A* were related to spastic ataxia (10/19 families, 52.6%); *KIF1A*, *POLG*, *SETX,* and *PNKP* to ataxia and neuropathy (4/19, 21.1%); *PNKP* to AOA (2/19, 10.5%); *HEXB* and *ATP1A3* to ataxia and dystonia (2/19, 10.5%); and *FA2H* to ataxia with cognitive impairment (1/19, 5.3%). The most frequent genes were those associated with spastic ataxia, ataxia, and neuropathy or AOA: *SACS* (4 families, 21.1%), *KIF1C* (2 families, 10.5%), and *PNKP* (3 families, 15.8%); the remaining genes were identified in single families only.

Variants were classified as pathogenic when having been previously reported as disease-causing (14/24), or by causing an early stop codon (nonsense and frameshift; 8/24) in genes where loss-of-function is a known disease mechanism (Appendix A). In addition, two variants were predicted to affect splicing by bioinformatics analysis (Appendix A)—c.3039-5T > G in *SPG11* and c.1166-2A > G in *KIF1C*—both in families with spastic ataxia; their detrimental effect on splicing was confirmed by minigene splicing-assays, concluding that both are expected to result in frameshifts (Figure 1; Appendix A).

We also identified two novel missense variants (Figure 2 and Appendix A), both in heterozygosity, classified as likely pathogenic (2/24): one in *CACNA1A* (c.4996C > G; p.Arg1666Gly; family AR267), at an amino-acid residue where a different missense change has been determined to be pathogenic, located in a well-established functional (transmembrane) domain (Appendix A) and not found in population databases; and one in *ATP1A3* (c.374T > A; p.Val125Glu; family AR278), a gene with low rate of benign missense variation, not reported in population databases, located in a mutation hotspot within a transmembrane domain (Appendix A) and predicted to be deleterious. Moreover, the WT residues of human *CACNA1A* (Arg1666) and ATP1A3 (Val125) are conserved across species, while the mutated residues are predicted to change their interactions, probably affecting protein structure (Figure 2). Besides the apparent AR inheritance, the variant in *ATP1A3* was confirmed to occur de novo (absent in the unaffected parents, family AR78). Unfortunately, we did not have parental samples to test for the *CACNA1A* variant (family AR267), but this variant probably occurs de novo, as these are a recurrent cause of *CACNA1A*-related ataxia [33]. In addition, we also identified one de novo variant in *KIF1A* (c.761G > A, described as pathogenic) in a patient with spastic ataxia, and confirmed the unaffected parents and sibling did not carry it (family AR49).

Thus, we reported 10 novel disease-associated variants in nine families (Appendix A). All variants were absent from population databases or were present only in heterozygosity with very low MAF.

### 3.2. Clinical Characterization

Beyond the 20 patients tested, we had clinical information (but no DNA sample) from nine additional affected relatives of probands (detailed data Table 1).

#### 3.2.1. Spastic Ataxia

Spastic ataxia was the largest phenotypic group, comprising 10 families (12 patients). With the exception of AR120 (*SACS*) and AR267 (*CACNA1A*), all patients had onset in the first decade of life. Its presenting symptoms comprised ataxia (AR53 (*SACS*), AR120 (*SACS*), AR103 (*ANO10*), AR109 (*SYNE1*), AR267(*CACNA1A*)), delayed motor milestones (AR77 (*SACS*), AR252 (*SACS*), AR111 (*KIF1C*)), cognitive regression (AR108 (*SPG11*)), and upper limb tremors (AR96(*KIF1C*)); we highlight, due to its rarity, the presence of seizures in AR77 (*SACS*) and the spastic ataxia phenotype in AR267 (*CACNA1A*). Regarding imaging, notably, family AR96 (*KIF1C*) showed basal ganglia hypointensities on T2/FLAIR sequences and AR111 (*KIF1C*) had brainstem atrophy.

#### 3.2.2. Ataxia and Neuropathy

This group included four families (10 patients). Onset was in the first decade in AR49 (*KIF1A*) and AR92 (*PNKP*), second decade in AR4 (*SETX*), and third decade in AR126 (*POLG*). Ataxia was the presenting feature in all families, with neuropathy developing later. It is noteworthy that in AR4 (*SETX*) and AR92 (*PNKP*) none of the patients had oculomotor apraxia. All patients from AR4 presented diplopia and nystagmus, but none from AR92 displayed such features. The patient from AR49 (*KIF1A*) had an extensor plantar response in the absence of spasticity. AR126 had a *POLG*-related ataxia, with the typical optic atrophy, vertical gaze palsy, and epilepsy. One of the members from AR92 died at age 35, with 29 years of disease duration. Imaging was only available in two families (AR4, AR49), all displaying cerebellar atrophy.

#### 3.2.3. Ataxia and Oculomotor Apraxia

Both families (4 patients) had onset in the first decade of life, and disease-causing variants in *PNKP*. Ataxia was the first symptom in all patients. OMA, neuropathy, and obesity were present in all individuals, and dystonia was highly frequent. Cerebellar atrophy was identified on magnetic resonance imaging (MRI); when tested, total proteins/albumin level was decreased, and cholesterol increased. In AR117, death occurred between 37 and 55 years of age, with a mean disease duration of 37 years.

#### 3.2.4. Ataxia and Dystonia/Ataxia and Cognitive Impairment

AR2 (*HEXB*) and AR278 (*ATP1A3*), with one patient each, presented ataxia and dystonia. AR2 (*HEXB*) had adult-onset ataxia, very prominent oromandibular dystonia and muscle wasting, while AR278 (*ATP1A3*) had onset at age 11 years, with paroxysmal lower limbs dystonia induced by walking, and progressive ataxia four years later. Over the disease course, dystonia became permanent and generalized. Both AR2 and AR278 had cerebellar atrophy on MRI.

AR16 (*FA2H*) was the only family with ataxia and cognitive impairment as the major phenotype, in a patient with delayed motor and cognitive milestones, who later developed ataxia and seizures. Cortical and cerebellar atrophy was present on MRI, and increased latency was identified in visual evoked potentials.

## 4. Discussion

In this study, we performed WES on 30 individuals from 19 families with (apparently) recessive cerebellar ataxia, aiming at providing a conclusive genetic diagnosis for them. These families had been identified during our national population survey in Portugal [10], but remained for many years without a molecular diagnosis, after testing the most common genes. More recently, new causal-genes were identified in undiagnosed families, including *PNKP* in eight families with AOA4 [11], *MAG* in one family with AOA and neuropathy [12], and *DAB1* in three AD families with SCA37 [34].

In this report, pathogenic and likely-pathogenic variants in several genes were identified and clearly associated with HCA (Table 1 and Appendix A). Division of the cohort into phenotypic subgroups had been performed during the original survey, and we retained this classification for a better genotype–phenotype characterization, and for clinician guidance in everyday practice. All the causal genes now identified had previously been described within their respective phenotypic subgroup [35,36,37,38]. The most frequent were *SACS*, *KIF1C,* and *PNKP.* To note, Friedreich ataxia, AOA (AOA1 and 2) and L-2-hydroxyglutaric aciduria had been previously screened and found to be the most prevalent types of AR-HCA [10]; *KIF1C* and *PNKP* were described only a few years later to be the causative genes for SPAX2 and AOA4 [11,39]. Present data reinforce *PNKP*-related ataxia as one of the most prevalent AR-HCA in Portugal [11]. Both *PNKP* and *SETX* were universally associated with neuropathy, but not with AOA; the reason why AR4 had not been previously tested for *SETX*. This has been described in the meantime [40]. Oculomotor apraxia, thus, should now be regarded as a clinical sign with high specificity, but less sensitivity for AOA2 and AOA4. Additionally, we identified four novel disease-variants in *SACS* and two in *KIF1C*, broadening the genetic spectrum of ARSACS and SPAX2.

We also provided functional data to validate the detrimental effect of two new splice-site variants (Figure 1) that would have been otherwise classified as variants of unknown clinical significance. Using minigene assays, we inferred that *SPG11* c.3039-5T > G variant creates a new 3′ acceptor splice site, while *KIF1C* c.1166-2A > G abolished the canonical 3′ acceptor slice site. Both were predicted to modify the open reading-frame, resulting in a premature stop codon.

### 4.1. Heterozygous De Novo Variants

All families analyzed had been classified as AR-HCA, based either on the presence of affected relatives in the same generation with unaffected parents; absence of other affected relatives; or consanguinity. Nevertheless, de novo variants in heterozygosity were identified (or presumed) in three families with only one affected member (Figure 2), highlighting that various modes of inheritance should considered when analyzing sequencing data in such cases. We identified one previously reported de novo missense variant in *KIF1A* [41], and two novel variants in *CACNA1A* and *ATP1A3 were* classified as likely pathogenic. *KIF1A* has been associated with AR and de novo AD diseases [42]. Patients with de novo *KIF1A*-related variants have shown a complex phenotype, including developmental delay, ataxia, spastic paraplegia, and neuropathy, with onset in the first year of life [41,43,44]. Our patient from family AR49 had had cerebellar ataxia since age 5 years, no spasticity, but an extensor plantar response, and neuropathy. Age of onset and combination of neurological syndromes were similar to those observed in Friedreich ataxia, but this was not the case in other cases previously described [43,44]. Nevertheless, our patient raises the hypothesis that heterozygous variants in *KIF1A* should be considered in ‘Friedreich ataxia-like’ patients.

Opposed to *KIF1A*, *CACNA1A* and *ATP1A3* are known to cause only autosomal dominant disorders [45,46,47,48]. CAG repeats in the last exon of the longest isoform of *CACNA1A* cause SCA6, and missense mutations affecting the calcium channel-encoding sequence cause progressive cerebellar ataxia, familial hemiplegic migraine (FHM1), and episodic ataxia type 2 (EA2) [49]. A congenital form of cerebellar ataxia with cognitive impairment has also been recently described [50]. A spastic ataxic phenotype, as observed in family AR267, has only been described once [35], probably being a less frequent presentation in *CACNA1A*-related disorders. Interestingly, *CACNA1A* variants were one of the most prevalent causes of AD-HCA, after SCA3 and dentatorubral-pallidoluysian atrophy (DRPLA), in our population-based survey [10]. Variants in *ATP1A3* produce a wide spectrum of AD neurological and psychiatric disorders, ranging from infantile to adult onset. Cerebellar ataxia is classically present in CAPOS syndrome (cerebellar ataxia, areflexia, *pes cavus*, optic atrophy, and sensorineural hearing loss), but atypical phenotypes with ataxia (either relapsing, of acute onset or slowly progressive) have been increasingly recognized [51]. The proband of AR278 had paroxysmal dystonia induced by walking, resembling exercise-induced dyskinesia (PED), and, later, slowly progressive ataxia and generalized dystonia. A phenotype similar to PED has been reported in a family with four affected individuals, two of which also display cerebellar ataxia [52].

### 4.2. Molecular Mechanisms

HCA can be caused either by loss-of-function, gain-of function, or a dominant-negative effect, in a multitude of apparently unrelated genes. Generally, AR-HCA is associated with loss-of-function variants, whereas AD-HCA can be caused by a combination of gain and/or loss of function [7]. Our study highlights the diversity of cerebellar ataxia-related genes and phenotypes, possibly reflecting different disease mechanisms (Appendix A). A pivotal question that remains unanswered is why the Purkinje cells are particularly affected [6,7]. Most genes/proteins identified in our study do not show elevated cerebellar expression, compared to other brain regions (Appendix A), as per RNA and protein data from the Human Protein Atlas, and in agreement with current literature [7]. We reviewed the molecular mechanisms underlying the main clinical phenotypes (Appendix A) and discuss potential common disease-pathways in AR-HCA below.

#### 4.2.1. Spastic Ataxia

Spasticity and pyramidal signs are hallmarks of several cerebellar ataxias [53,54]. As demonstrated by the diversity of genes causing spastic ataxia in our study, there is not a particular molecular pathway underlying this phenotype. Nevertheless, we identified one molecular mechanism that may be shared between a set of genes: (1) *SACS*, *KIF1C*, *SPG11,* and *SYNE1* interact with the cytoskeleton and may function to ensure proper vesicle/organelle trafficking and/or neuronal structure.

*SACS* encodes the large protein sacsin, the function of which has not yet been clearly established. The presence of both ubiquitin-like (UBL) and DNaJ domain (implicated in chaperon-mediated folding process), and a potential role in the degradation of aberrant ataxin-1, suggest that sacsin may integrate the ubiquitin–proteasome system and Hsp70 chaperone function [55]. Moreover, studies in *SACS* knockout mice, revealed that sacsin regulates the neurofilament cytoskeleton and mitochondria dynamics [56,57]. Most *SACS* variants lead to the complete loss of sacsin, although some missense variants are associated with low levels of the protein [58,59]. Since most variants are upstream of the C-terminal DnaJ domain, it is speculated that ARSACS is associated with loss of chaperone function [60]. Concordantly, we identified known and novel variants in an unspecific region or within the sacsin-repeating region with homology with Hsp90 (Appendix A); all upstream of the DNaJ domain.

Kinesin family member 1C (*KIF1C*) is a ubiquitously expressed motor protein involved in both anterograde transport of vesicles and retrograde transport from Golgi to the ER [61]. Few reports have analyzed the impact of *KIF1C* variants, but it was suggested that variants in the motor domain may affect microtubule binding and impair ATP hydrolysis, while truncating variants lead to reduced protein levels and may impair cargo binding [62]. Therefore, truncating variants, such as the ones identified in our study, are probably associated with a loss-of-function mechanism.

*SPG11* encodes spatacsin, a large protein involved with neuronal axonal growth, intracellular cargo trafficking, and lysosome function. SPG11 variants span the entire spatacsin protein [63]. Functional studies of truncating variants indicated that spatacsin loss-of-function is at the basis of neurodegeneration, as evidenced by reduced expression of spatacsin [64]; reduction in the anterograde vesicle trafficking, indicative of impaired axonal transport [65]; reduction in axonal complexity and neurite outgrowth [65]; or the presence of abnormal lysosomes [64]. Given the nature of the variants we identified, it is plausible that they also cause loss of spatacsin function.

*SYNE1* is a large gene that encodes the synaptic nuclear envelope protein 1 (SYNE1), also known as nuclear envelope spectrin 1 (nesprin 1). This protein mediates the link from the nuclear envelope to the actin cytoskeleton (LINC complex), being important for nuclear migration [66,67]. Most *SYNE1* variants associated with ARCA1 are nonsense or frameshift, and span the giant nesprin-1 isoform, but exclude the C-terminal Klarsicht-ANC-Syne-homology (KASH) [68]. Thus, ARCA1 may be caused by defective synaptic transmission through synaptic vesicles trafficking impairment and/or cerebellar dendrites defects; however, this remains to be confirmed [69]. Nevertheless, many missense variants were also identified in ARCA1, including one identified by us, within a spectrin repeat domain.

In addition, variants in *ANO10*, which encodes a putative calcium-activated chloride channel, named anoctamin-10 (ANO10), are associated with SCAR10/ARCA3 [70], with decreased coenzyme Q10 (CoQ10) levels. One study showed that truncating variants cause a reduction in *ANO10* mRNA expression in patient’s skin fibroblasts [71], but the pathogenic mechanism of *ANO10* variants remains unexplored.

#### 4.2.2. Ataxia and Neuropathy

Peripheral neuropathy is a common feature associated to HCAs; but, as far as we know, this combination has not been linked with specific disease pathways. In this study, we found three genes (*POLG*, *SETX,* and *PNKP*) with various molecular functions, causing recessive ataxia and neuropathy.

*POLG* encodes the catalytic subunit of the mitochondrial DNA polymerase gamma protein (POLG), which functions in the replication and repair of mtDNA [72]. Functional studies showed that missense variants cause decreased activity, DNA binding and processivity of the polymerase, and reduction of functional mtDNA [73,74].The *POLG* variant identified in our study (p.Trp748Ser) is within the C-terminal domain of the polymerase, an important domain for its activity [75]. We can, therefore, speculate that it causes reduction or loss of POLG function.

*SETX* and *PNKP* are discussed in the following sub-section.

#### 4.2.3. Ataxia and Oculomotor Apraxia

AOA has been associated with the DNA damage repair pathway. Ataxia telangiectasia, AOA1, AOA2, AOA4, *XRCC1*-AOA, and spinocerebellar ataxia with axonal neuropathy 1 are all caused by genes with a role in DNA repair [9]. We identified variants in *SETX* and *PNKP,* causing AOA2 and AOA4, respectively. Interestingly, in one family with pathogenic variants in *SETX* and one in *PNKP*, oculomotor apraxia was not present. Both had been phenotypically classified as ‘ataxia with neuropathy’, due to severe neuropathy. This possible occurrence has been previously described and is reinforced in this study.

*SETX* encodes a large nuclear protein termed senataxin, with an RNA/DNA helicase domain. Senataxin is involved in RNA metabolism, DNA maintenance, and damage response. Several types of biallelic variants in *SETX* were identified in AOA2 [76,77,78], probably causing a loss-of-function phenotype. In addition, dominant missense variants in *SETX* cause amyotrophic lateral sclerosis (ALS) with juvenile onset (ALS4) [79], suggesting that gain-of-function leads to ALS4. Altered gene expression and mRNA processing, and increased susceptibility to oxidative DNA damage, have all been associated with *SETX* variants [79,80,81,82]. There have been no functional studies of the SETX p.Gly2047Cys variant [83], but as it lies within the DNA/RNA helicase domain, it may cause a reduction or loss of senataxin function.

Variants in *PNKP*, encoding the nuclear polynucleotide kinase 3′ phosphatase protein (*PNKP*), result in a range of AR disorders. The enzymatic activity of *PNKP* is involved in the repair of both DNA double strand-break (DSB) and single-strand break (SSB); impaired function can cause neurodevelopmental dysfunction (microcephaly, seizures, and developmental delay (MCSZ)) [84] and neurodegeneration (AOA4 and Charcot-Marie-Tooth disease (CMTD)) [11,36]. Few functional studies have described the effects of *PKNP* variants. Nevertheless, most variants seem to cause reduced stability and levels of *PNKP*, and reduced DNA phosphatase activity in MCSZ or reduced kinase activity in neurodegenerative diseases. These ultimately lead to reduced DNA repair [85,86]. Moreover, several types of variants were identified in *PNKP*; most are in the kinase domain, particularly in neurodegenerative diseases [11,87], as shown in this study.

#### 4.2.4. Ataxia and Cognitive Impairment or Dystonia

Additionally, other neurological signs may be present in AR-HCA, such as dystonia, and cognitive impairment. In our study, we identified a known variant in *HEXB* in one family with Sandhoff disease, and a new variant in *FA2H* causing ataxia with cognitive impairment.

Sandhoff disease is caused by variants in *HEXB* encoding the β subunit of the enzyme hexosaminidase (HEXB) [88]. HEXB is a lysosomal hydrolase that catalyzes the degradation of GM2 ganglioside. The spectrum of *HEXB* variants is wide, but most reduce GM2 hydrolysis, resulting in the accumulation of GM2 ganglioside [89,90]. Moreover, it was reported that several variants, including missense, lead to reduced mRNA expression and affect HEXB protein structure [91,92,93]. Particularly, the p.Arg505Gln variant, located in the catalytic domain, was reported to disturb the biochemical properties and cause loss of activity of HEXB and accumulation of GM2 ganglioside [90,94].

*FA2H* encodes the fatty acid 2-hydroxylase (FA2H) protein, which is involved in the synthesis of 2-hydroxy fatty acid galactolipids, the most abundant lipids in the myelin sheath [95]. Pathogenic variants in *FA2H* are speculated to cause neurodegeneration through a loss-of-function mechanism, since there is evidence of decreased hydroxylation of myelin lipids [96]. Moreover, a missense variant causing neurodegeneration with brain iron accumulation, led to reduced FA2H protein expression [97]. Since the *FA2H* variant we identified is predicted to cause a frameshift, it is also possible it leads to loss-of-function.

## 5. Conclusions

Our study provided the molecular diagnosis of 19 families with various types of HCA through WES, expanding the genetic and phenotypic spectrum of HCA. Most of the mutated genes caused a spastic ataxia phenotype, but *PNKP* associated with either ataxia and neuropathy or AOA was also amongst the most frequent causal-genes. We highlight three families with de novo variants in *KIF1A*, *CACNA1A,* and *ATP1A3*, despite having been initially classified has AR. These results evidence the importance of performing a differential diagnosis (AR vs. AD forms) in the absence of affected relatives of the proband. All the remaining families were associated with AR-HCA related genes. In addition, we provided a review on potential common mechanisms underlying neurodegeneration in AR-HCA, namely cytoskeleton function in spastic ataxia, and DNA repair in AOA. Translation of genetic findings into a better understanding of HCA mechanisms may help the development of effective therapies [6,7].

## Figures and Tables

**Figure 1 cells-11-00981-f001:**
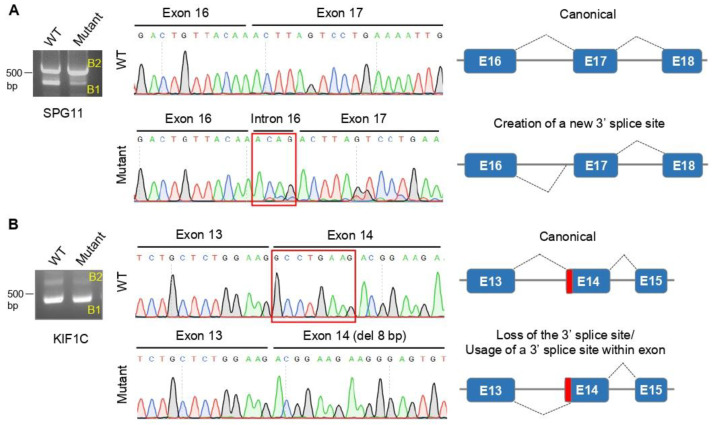
Analysis of *SPG11* (c.3039-5T > G) (**A**) and *KIF1C* (c.1166-2A > G) (**B**) splice-site variants, with minigene splicing assays. Agarose gel electrophoresis of the transcripts generated by the wild-type (WT) and mutant minigene constructs (on the left): B1 bands (<500 bp) represent transcripts affected by the variants; B2 bands (>500 bp) represent amplified transcripts with intron retention, not affected by the variant. Sanger sequencing of the transcripts corresponding to B1 bands (in the middle). Schematic representation of resulting splicing events (on the right): (**A**) *SPG11* variant creates a new 3′ acceptor splice site, resulting in the retention of 4 nucleotides of intron 16; (**B**) *KIF1C* variant abolished the 3′ acceptor splice-site, leading to the usage of a splice site within exon 14 and resulting in the deletion of 8 nucleotides from exon 14. Electropherograms showing the location of the splice-site variants on genomic DNA are shown in Appendix A.

**Figure 2 cells-11-00981-f002:**
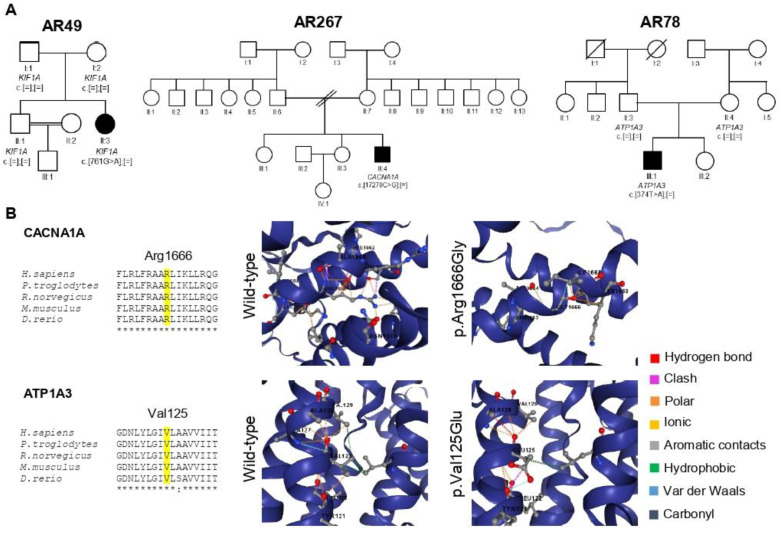
Pedigrees of families harboring de novo variants (**A**), and conservation of the new missense variants and protein models of CACNA1A and ATP1A3 (**B**). (**A**) Black symbols represent individuals with cerebellar ataxia. (**B**) Sequence alignment of the residues surrounding the mutated residues in CACNA1A and ATP1A3 against other species was performed using the Clustal Omega program. Protein models of CACNA1A and ATP1A3 showing altered residues interactions upon amino acid changes (performed using DynaMut2). The Arg1666 and Val135 residues in CACNA1A and ATP1A3, respectively, are in the center, displaying the various interactions with nearby residues in different colors.

**Table 1 cells-11-00981-t001:** Clinical characterization of the families.

Family	Ind. Code (Gender)	Tested	Gene	Disease Causing Variants	Consanguinity	Age				First Symptom	Ataxia	OMA	Diplopia	Pyramidal Signs	Peripheral Neuropathy	Dystonia	Cognitive Impairment	Other Symptoms/Signs	Imaging	Others
						Onset	Obs.	Wheelchair	Death											
**Spastic ataxia**																				
**AR53**	IV.1 (F)	Yes	*SACS*	c.8148del; c.8148del	Yes	6	53	47	NA	Ataxia	Yes	No	No	Yes	No	No	No		NAv	
**AR77**	II.3 (M)	Yes	*SACS*	c.1373C >T; c. 5440_5449del	No	<1	31	NA	NA	Delayed motor milestones	Yes	No	No	Yes	Yes	No	No	Seizures 1-16y Nystagmus	MRI: cerebellar atrophy	
**AR120**	II.1 (F)	Yes	*SACS*	c.3066del; c.2656_2666del	No	15	66	29	NA	Ataxia	Yes	No	No	Yes	Yes	No	No	Nystagmus	NAv	
**AR252**	II.1 (M)	Yes	*SACS*	c.814C > T; c.814C > T	Not clear	<1	57	54	NA	Delayed motor milestones	Yes	No	No	Yes	Yes	No	No	Kyphoscoliosis	NAv	
**AR96**	IV.1 (M)	Yes	*KIF1C*	c.393_396del; c.393_396del	Yes	9	52	41	NA	UL tremor	Yes	No	No	Yes	Yes	No	Mild deterioration	Nystagmus	MRI: pallidum, putamen, thalamus hypointensities (T2/FLAIR)	
	IV.2 (F)	Yes				8	50	44	NA	UL tremor	Yes	No	No	Yes	Yes	No	Mild deterioration	Nystagmus	MRI: vermis atrophy	
**AR111**	III.3 (F)	Yes	*KIF1C*	c.1166-2A > G; c.1166-2A > G	Yes	<1	41	NA	NA	Delayed motor milestones	Yes	No	No	Yes	Yes	No	No	Vertical gaze restriction Mild dysmorphism Pes cavus	CT: brainstem atrophy	
**AR103**	III.3 (F)	Yes	*ANO10*	c.132dup; c.132dup	Yes	7	40	NA	NA	Ataxia	Yes	No	Yes	Yes	No	No	No	Nystagmus Migraine	MRI: cerebellar atrophy	
**AR108**	II.3 (F)	Yes	*SPG11*	c.1951C > T; c.3039-5T > G	No	7	43	NA	NA	Cognitive regression	Yes	No	No	Yes	No	No	Yes		MRI: Thin corpus callosum	
	II.2 (M)	No				8	47	NA	NA	Cognitive regression	Yes	No	No	Yes	No	No	Yes		MRI: Chiari type1	
**AR109**	III.2 (M)	Yes	*SYNE1*	c.17270C > G; c.17270C > G	Not clear	NAv	67	NA	NA	Ataxia	Yes	No	No	Yes	No	No	No		NAv	
**AR267**	III.4 (M)	Yes	*CACNA1A*	c.4996C > G	No	19	21	NA	NA	Ataxia	Yes	No	No	Yes	No	No	Yes		MRI: cerebellar atrophy	
**Ataxia and neuropathy**																				
**AR4**	VI.27 (M)	Yes	*SETX*	c.1514G > A; c.1514G > A	Yes	14	37	NA	NA	Ataxia	Yes	No *	Yes	No	Yes	No		Nystagmus	CT: cerebellar atrophy	
	VI.7 (F)	No				13	49	NA	NA	Ataxia	Yes	No *	Yes	No	Yes	No		Nystagmus	CT: cerebellar atrophy	
	VI.10 (F)	No				11	43	NA	NA	Ataxia	Yes	No *	Yes	No	Yes	No		Nystagmus	CT: cerebellar atrophy	
	VI.19 (F)	No				12	43	NA	NA	Ataxia	Yes	No *	Yes	No	Yes	No		Nystagmus Obesity	CT: cerebellar atrophy	
	VI.20 (F)	No				13	42	NA	NA	Ataxia	Yes	No *	Yes	No	Yes	No	Mild retardation	Nystagmus	CT: cerebellar atrophy	
	VI.31 (F)	No				11	32	NA	NA	Ataxia	Yes	No *	Yes	No	Yes	No		Nystagmus	CT: cerebellar atrophy	
**AR49**	II.3 (F)	Yes	*KIF1A*	c.761G > A	No	5	26	NA	NA	Ataxia	Yes	No	NAv	Yes	Yes	No	Mild retardation		CT: cortical cerebellar atrophy	
**AR92**	II.1 (M)	Yes	*PNKP*	c.1123G > T; c.1123G > T	Not clear	6	33	16	35	Ataxia	Yes	No	No	No	Yes	No	No		NAv	
	II.4 (M)	No				7	27	16	NA	Ataxia	Yes	No	No	No	Yes	No	No		NAv	αFP: N
**AR126**	V.7 (F)	Yes	*POLG*	c.2243G > C; c.2243G > C	Yes	25	41	NA	NA	Ataxia	Yes	No	No	No	Yes	No	Yes	Optic atrophy Vertical gaze palsy Nystagmus Epilepsy	NAv	
**Ataxia and oculomotor apraxia**																				
**AR86**	II.2 (F)	Yes	*PNKP*	c.1123G > T; c.1253_1269dup	No	8	17	NA	NA	Ataxia	Yes	Yes	No	No	Yes	No	Yes	Nystagmus Obesity	MRI: cerebellar atrophy (++vermis)	
**AR117**	II.1 (M)	Yes	*PNKP*	c.1221_1223del; c.1123G > T	Yes	8	32	16	37	Ataxia	Yes	Yes	NAv	No	Yes	Yes (UL)	No	Obesity	NAv	
	II.2 (F)	No				7	35	15	42	Ataxia	Yes	Yes	NAv	No	Yes	Yes (UL)	No	Obesity	MRI: cerebellar atrophy	αFP: ↑ Proteins: ↓ Albumin: ↓ Cholesterol: ↑
	II.3 (F)	No				7	29	15	55	Ataxia	Yes	Yes	NAv	No	Yes	Yes (UL)	No	Obesity	MRI: cerebellar atrophy	Proteins: ↓ Albumin: ↓ Cholesterol: ↑
**Ataxia and dystonia**																				
**AR2**	IV.1 (M)	Yes	*HEXB*	c.1514G > A; c.1514G > A	Yes	24	51	NA	NA	Ataxia	Yes	No	No	Yes	No	Yes (OM)	No	Nystagmus Muscle atrophy ↓ vibration hallux	MRI: cerebellar, atrophy	
**AR278**	III.1 (M)	Yes	*ATP1A3*	c.374T > A	No	11	44	NA	NA	Parox dystonia	Yes	No	No	No	No	Yes	Yes	Supranuclear vertical gaze palsy	MRI: cerebellar atrophy	
**Ataxia and cognitive impairment**																				
**AR16**	V.1 (F)	Yes	*FA2H*	c.619_620del; c.619_620del	Yes	<1	14	NA	NA	Delayed motor/cognitive milestones	Yes	No	No	Yes	No	No	Yes	Seizures, Aggressive behavior	MRI: cerebellar atrophy, mild cortical atrophy	VEP: ↑ BEP, SSEP: N

αFP—α feto-protein; BEP—Brainstem evoked potentials; CT—Computed tomography; F—Female; Ind—Individual; M—Male; MRI—Magnetic resonance imaging; NA—Not applicable; Nav—Not available; Obs—Observation; OM—Oromandibular; OMA—Oculomotor apraxia; Parox—paroxysmal; UL—Upper limbs; SSEP—Somatosensory evoked potentials; VEP—Visual evoked potentials; ↓—decreased; ↑—increased * No clear oculomotor apraxia, but all had a poverty of spontaneous saccadic eye movements.

## Data Availability

Not applicable.

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
