# Peer review of "Molecular Characterization of Portuguese Patients with Hereditary Cerebellar Ataxia"

_cells, 2022, doi:10.3390/cells11060981_

Round 1
Reviewer 1 Report
Santos et al. wrote a very interesting manuscript entitled "Molecular characterization of Portuguese patients with recessive cerebellar ataxia ". The authors evaluated 19 families with hereditary cerebellar ataxia (HCA), presumed ARCA (autosomal recessive cerebellar ataxia), performed whole-exome-sequencing (WES). In conclusion, the authors stated that they provide a genetic diagnosis for HCA families (including cases of ARCA and sporadic cases with autosomal dominant cerebellar ataxias - ADCA or Spinocerebellar ataxias- SCAs). I suggest to the authors a new title " Molecular characterization of Portuguese patients with hereditary cerebellar ataxias", because there are families with ARCA (predominantly) but 3 families had autosomal dominant cerebellar ataxias- ADCA or SCAs !.
Author Response
Reviewer 1
Santos et al. wrote a very interesting manuscript entitled "Molecular characterization of Portuguese patients with recessive cerebellar ataxia ". The authors evaluated 19 families with hereditary cerebellar ataxia (HCA), presumed ARCA (autosomal recessive cerebellar ataxia), performed whole-exome-sequencing (WES). In conclusion, the authors stated that they provide a genetic diagnosis for HCA families (including cases of ARCA and sporadic cases with autosomal dominant cerebellar ataxias - ADCA or Spinocerebellar ataxias- SCAs). I suggest to the authors a new title " Molecular characterization of Portuguese patients with hereditary cerebellar ataxias", because there are families with ARCA (predominantly) but 3 families had autosomal dominant cerebellar ataxias- ADCA or SCAs.
Response: We would like to thank the Reviewer for the positive comment and suggestion. We agreed and changed the manuscript title to “Molecular characterization of Portuguese patients with hereditary cerebellar ataxia”.
Reviewer 2 Report
This article by Santos and Damasio et al. discusses results from a whole-exome sequencing study of individuals from families affected by presumed autosomal recessive cerebellar ataxia, for which disease-causing genes had not been identified. The authors identified causal genes for 19 families, which included 10 novel gene variants across 9 families. They additionally characterize two point mutations that putatively alter the splice sites for SPG11 and KIF1C, along with identifying three de novo mutations causing autosomal-dominant cerebellar ataxia. The authors also provide a detailed discussion of clinical and genetic features of autosomal recessive cerebellar ataxia involving the genes they identified in this study.
Overall, the study is mostly descriptive of genes already known to cause recessive cerebellar ataxia, but it is useful to investigate how frequently gene variants appear in genetic screens of ataxia. The whole exome sequencing results are presented clearly and with comprehensive discussion. I have minor comments about the discussion and presentation of data in this study.
- The identified KIF1A, CACNA1A, and ATP1A3 mutations are thought to be de novo mutations resulting in autosomal dominant cerebellar ataxia. However, these genes are included in the discussion sections that speculate about molecular mechanisms of autosomal recessive cerebellar ataxia. These genes are already discussed in the section discussing heterozygous de novo variants and I believe should not be used to draw strong conclusions about molecular mechanisms in each group of AR ataxias, since their status as AD is more likely.
- With CACNA1A causing AD cerebellar ataxia, and ANO10 not having a well-characterized role in calcium handling as a calcium-activated chloride channel, I do not feel that the speculation about calcium homeostasis as a major mechanism of autosomal recessive ataxia is strongly supported. There is a large body of evidence that calcium homeostasis is a major disease mechanism in polyglutamine spinocerebellar ataxia, but unless there are other causative genes for recessive ataxia involved in calcium homeostasis, I believe that these statements should be omitted from the abstract and discussion.
- How many families were initially screened in the WES study? Were all 83 patients described in the methods section included? It would be helpful to hear what percentage of patients had causative genes identified, and how many were still unable to achieve a genetic diagnosis after the WES study (i.e. the yield of the WES).
Minor formatting:
- In Figure 1, it would be helpful to highlight where the splice site mutations occur in the Sanger sequencing results. Perhaps a box around the bases that are altered, or underlined. It is difficult for me to easily identify what has changed.
- Since Table 1 appears across several pages, the header column should appear on every page. Otherwise it is difficult for the reader to know what each column of the table represents.
Author Response
Reviewer 2
This article by Santos and Damasio et al. discusses results from a whole-exome sequencing study of individuals from families affected by presumed autosomal recessive cerebellar ataxia, for which disease-causing genes had not been identified. The authors identified causal genes for 19 families, which included 10 novel gene variants across 9 families. They additionally characterize two point mutations that putatively alter the splice sites for SPG11 and KIF1C, along with identifying three de novo mutations causing autosomal-dominant cerebellar ataxia. The authors also provide a detailed discussion of clinical and genetic features of autosomal recessive cerebellar ataxia involving the genes they identified in this study.
Overall, the study is mostly descriptive of genes already known to cause recessive cerebellar ataxia, but it is useful to investigate how frequently gene variants appear in genetic screens of ataxia. The whole exome sequencing results are presented clearly and with comprehensive discussion. I have minor comments about the discussion and presentation of data in this study.
- The identified KIF1A, CACNA1A, and ATP1A3 mutations are thought to be de novo mutations resulting in autosomal dominant cerebellar ataxia. However, these genes are included in the discussion sections that speculate about molecular mechanisms of autosomal recessive cerebellar ataxia. These genes are already discussed in the section discussing heterozygous de novo variants and I believe should not be used to draw strong conclusions about molecular mechanisms in each group of AR ataxias, since their status as AD is more likely.
Response: We are very grateful to the Reviewer for the pertinent comments and suggestions.
As suggested, we removed from the discussion, and from Table S2 supporting information, the molecular mechanisms of KIF1A, CACNA1A and ATP1A3, focusing the discussion only on genes causing recessive ataxia.
- With CACNA1A causing AD cerebellar ataxia, and ANO10 not having a well-characterized role in calcium handling as a calcium-activated chloride channel, I do not feel that the speculation about calcium homeostasis as a major mechanism of autosomal recessive ataxia is strongly supported. There is a large body of evidence that calcium homeostasis is a major disease mechanism in polyglutamine spinocerebellar ataxia, but unless there are other causative genes for recessive ataxia involved in calcium homeostasis, I believe that these statements should be omitted from the abstract and discussion.
Response: Some forms of recessive cerebellar ataxia are associated with calcium homeostasis dysregulation [AFG3L2 (SPAX5), ANO10, CA8 (congenital cerebellar ataxia with mental retardation, ITPR1 (cases of recessive SCA29)], but not as clearly as in polyglutamine spinocerebellar ataxia. We agreed with the reviewer; thus, as suggested, we removed the statement about calcium homeostasis from the abstract and discussion.
- How many families were initially screened in the WES study? Were all 83 patients described in the methods section included? It would be helpful to hear what percentage of patients had causative genes identified, and how many were still unable to achieve a genetic diagnosis after the WES study (i.e. the yield of the WES).
Response: From the 61 undiagnosed families (83 patients) on the initial study, we only had genomic DNA available from 27 families. We performed WES on at least one member of these 27 families and reached a conclusive diagnosis in 19 families, which are presented in this study. Thus, the yield of WES on our study would be of 70.37%. We believe that this high yield results from identification of the most frequent genes in the initial study and also because we were not able to test the remaining 34 families with no DNA available. For that, we are not comfortable to take conclusions on the yield of WES. We add the following sentence on the methods:
Lines 93-94 (methods): “From the 61 undiagnosed families, we only had genomic DNA available of 27 families, with a final diagnosis being reached in the 19 here described.”
Minor formatting:
- In Figure 1, it would be helpful to highlight where the splice site mutations occur in the Sanger sequencing results. Perhaps a box around the bases that are altered, or underlined. It is difficult for me to easily identify what has changed.
Response: As suggested, we changed figure 1 to highlight the differences in the transcripts (cDNA nucleotide sequence) and updated the legend. Also, we add Figure 1 supporting information to show the splice site mutations on Sanger sequencing and its localization within introns.
- Since Table 1 appears across several pages, the header column should appear on every page. Otherwise it is difficult for the reader to know what each column of the table represents.
Response: We agreed and changed the table accordingly.